# Creating an Implementation Enhancement Plan for a Digital Patient Fall Prevention Platform Using the CFIR-ERIC Approach: A Qualitative Study

**DOI:** 10.3390/ijerph20053794

**Published:** 2023-02-21

**Authors:** Alana Delaforce, Jane Li, Melisa Grujovski, Joy Parkinson, Paula Richards, Michael Fahy, Norman Good, Rajiv Jayasena

**Affiliations:** 1Australian e-Health Research Centre, Health and Biosecurity, Commonwealth Scientific and Industrial Research Organisation, Herston, QLD 4029, Australia; 2Australian e-Health Research Centre, Health and Biosecurity, Commonwealth Scientific and Industrial Research Organisation, Westmead, NSW 2145, Australia; 3Nursing and Midwifery Services, Maitland Hospital, Hunter New England Local Health District, Maitland, NSW 2323, Australia; 4Australian e-Health Research Centre, Health and Biosecurity, Commonwealth Scientific and Industrial Research Organisation, Parkville, VIC 3052, Australia

**Keywords:** fall prevention, Consolidated Framework for Implementation Research, Expert Recommendations for Implementing Change, digital workflow, platform, enhancement plan

## Abstract

(1) Background: Inpatient falls are a major cause of hospital-acquired complications (HAC) and inpatient harm. Interventions to prevent falls exist, but it is unclear which are most effective and what implementation strategies best support their use. This study uses existing implementation theory to develop an implementation enhancement plan to improve the uptake of a digital fall prevention workflow. (2) Methods: A qualitative approach using focus groups/interview included 12 participants across four inpatient wards, from a newly built, 300-bed rural referral hospital. Interviews were coded to the Consolidated Framework for Implementation Research (CFIR) and then converted to barrier and enabler statements using consensus agreement. Barriers and enablers were mapped to the Expert Recommendations for Implementing Change (ERIC) tool to develop an implementation enhancement plan. (3) Results: The most prevalent CFIR enablers included: relative advantage (*n* = 12), access to knowledge and information (*n* = 11), leadership engagement (*n* = 9), patient needs and resources (*n* = 8), cosmopolitanism (*n* = 5), knowledge and beliefs about the intervention (*n* = 5), self-efficacy (*n* = 5) and formally appointed internal implementation leaders (*n* = 5). Commonly mentioned CFIR barriers included: access to knowledge and information (*n* = 11), available resources (*n* = 8), compatibility (*n* = 8), patient needs and resources (*n* = 8), design quality and packaging (*n* = 10), adaptability (*n* = 7) and executing (*n* = 7). After mapping the CFIR enablers and barriers to the ERIC tool, six clusters of interventions were revealed: train and educate stakeholders, utilize financial strategies, adapt and tailor to context, engage consumers, use evaluative and iterative strategies and develop stakeholder interrelations. (4) Conclusions: The enablers and barriers identified are similar to those described in the literature. Given there is close agreement between the ERIC consensus framework recommendations and the evidence, this approach will likely assist in enhancing the implementation of Rauland’s Concentric Care fall prevention platform and other similar workflow technologies that have the potential to disrupt team and organisational routines. The results of this study will provide a blueprint to enhance implementation that will be tested for effectiveness at a later stage.

## 1. Introduction

Hospital inpatient falls are a persistent problem both internationally and in Australia. In 2020–2021, Australian hospitals reported more than 47,000 falls [1,2], with just over 3000 of those resulting in fractures or intracranial injuries. Patients with these types of injuries can remain in the hospital for much longer (on average, 18.8 days), leading to higher hospital-acquired morbidity and mortality [3]. In addition to patient harm, the cost of falls is significant. Based on an average acute overnight hospital stay cost of approximately AUD 2074, falls can be associated with more than AUD 38,991 in additional costs per patient or in the range of AUD 117 million additional costs for hospitals annually [1,3].

Fall prevention interventions exist, such as exercise, medication review, environmental or assistive technologies (including bed or chair alarms), social environment interventions that target staff members, changes in the organisational system and knowledge interventions [4]. However, it remains unclear which interventions are most effective. A 2018 Cochrane review by Cameron et al. found that although there is some evidence to support the effect of fall prevention interventions, it is not of good quality or sufficiently generalizable [4]. The results of the Cochrane systematic review show that many of the studies available focused on a single site only and delivered interventions in different combinations, making it impossible to clearly identify what approach helps prevent hospital inpatient falls. Another systematic review in 2022 supported the findings of the Cochrane review, although additionally found that patient and staff education is associated with a reduction in fall rates (RaR = 0.70 [0.51–0.96], *p* = 0.03) among high quality studies. Further research is needed on a wider range of fall prevention interventions to determine which are most effective [5].

Additionally, which implementation strategies are most effective at supporting fall prevention intervention uptake remains unclear [6]. Hempel et al.’s systematic review found that while some fall prevention implementation strategies appear promising, it was not possible to make firm generalisations about their effect due to suboptimal reporting [7]. Many included studies did not clearly report their approach, the intervention, the context or the study cohort differences (e.g., before and after groups not adequately described) [7]. Research is needed to address the gap in understanding what implementation strategies best support the implementation of fall prevention interventions.

Implementation science offers a range of theories and frameworks that help identify factors that influence the uptake of evidence-based practice [8]. In this study, the Consolidated Framework for Implementation Research (CFIR) is used to identify barriers and enablers [9]. The CFIR is the result of a systematic review of implementation frameworks and provides a shared taxonomy of constructs that influence implementation at multiple levels (e.g., innovation, organisation, process, external and individual) [9]. Since the development of the CFIR, the Expert Recommendations for Implementing Chance (ERIC) tool has been created to facilitate the selection of theoretically informed strategies to address CFIR barriers [10]. The ERIC tool was developed based on the consensus of implementation experts as to what strategies are most effective in addressing CFIR barriers [10]. The ERIC tool has since undergone refinements based on further research [11,12,13].

The ERIC tool can be accessed through the website: Strategy Design—The Consolidated Framework for Implementation Research (cfirguide.org, accessed on 1 February 2023) and suggests level one and level two strategies based on the level of consensus achieved. A level one strategy had greater than 50% agreement by experts that they are effective in addressing a given CFIR construct, and level two had less than 50%. Since its development, this work has been refined to assist with focusing on improvement efforts. Waltz and colleagues undertook a hierarchical cluster analysis using the consensus of implementation experts to help group ERIC strategies into more focused themes [12]. This study leverages the suggestions of Waltz et al. and clusters recommended strategies on this basis. Qualitative methods are used to identify enablers and barriers to the implementation of the Rauland Concentric Care fall prevention platform and then mapped across to a consensus-derived framework to identify theory-informed strategies that should help address them. The results of this study will provide a blueprint to enhance implementation that will be tested for effectiveness at a later stage.

## 2. Materials and Methods

**Aim**: This study aimed to develop an implementation enhancement plan by identifying enablers and barriers to using the Rauland Concentric Care fall prevention platform and selecting theoretically informed strategies to address them.

**Design**: Using a qualitative approach, we sought the perspectives of hospital staff who may influence the implementation and operationalisation of the Rauland Concentric Care fall prevention platform.

**Setting**: The setting of this study was four inpatient wards within a newly built, 300-bed rural referral hospital that provides medical, surgical and maternity services to approximately 92,000 public patients annually. In early 2022, hospital staff moved from an older campus to the new one, where the Concentric Care fall prevention platform was installed.

**Clinical innovation description**: The Concentric Care fall prevention platform is a technology solution that (as currently available) consists of a speech-enabled nurse call communication system, location services engine and digital dashboards. Specific features of the innovation platform include nurse call buttons on the bedside handsets with two-way communication capability, audio/microphone stations in bathrooms, Wi-Fi enabled clinical mobile devices, staff station consoles and smart bed integration. The Concentric Care fall prevention system provides a solution for new workflows to be implemented to assist with providing care to high-risk fall patients while leveraging the benefits of the Concentric Care technology (see Appendix A). Education in relation to the new Concentric Care fall prevention platform was available several months before the site went live via mandatory online training and this was complimented by face-to-face education provided to staff by clinical experts just before the site went live. Education is also provided to patients, who are asked to watch an induction video from Rauland via the patient entertainment system that provides guidance and orientation of expected fall prevention protocols.

**Participants and recruitment**: Granting of full ethical approval occurred before the commencement of the study by the district-level Human Research Ethics Committee (HREC) (reference: 2021/ETH11953), and reciprocal ethics approval was granted from CSIRO Health and Medical Human Ethics Committee (reference: 2022_012_RR).

Approximately six months after initial implementation, nurses, nurse managers and nurse educators were able to participate if they were involved with the implementation of the Concentric Care fall prevention platform or worked in a ward in which the system was in use. Eligible staff were invited by the onsite study representative known to them (MG). Invitations were made verbally or through email, following the provision of staff details from nurse unit managers who identified staff that work on ward/s using the Concentric Care fall prevention platform. Once participants completed and signed the information and consent form, a member of the research team (JL) contacted the onsite contact (MG) to arrange a time and space to hold virtual focus group/interview. Focus groups/interview sessions were scheduled by the study representative at the hospital based on the availabilities of participants. Limited characteristics of participants are reported in the results due to the small sample size.

**Data collection**: A total of 12 participants were included in the interview/focus groups, which were informed using the CFIR interview guide (see Appendix A) [9]. Focus groups were held across five sessions with eleven participants. One individual interview was undertaken with a senior member of staff who was not available to participate in the focus groups. Interview/focus groups of 1–3 participants of 45–60 min in length were conducted by two team members (AD, JL) and recorded virtually using Webex™. A quiet and private room was used for the interview/focus groups. Where multiple people were included in the focus group, the interviewers directed questions at specific participants to encourage responses and equal participation. Audio recordings were professionally transcribed and the data were de-identified.

**Data analysis**: Two researchers (AD, JL) independently coded each transcript with NVivo (QSR International) using the Consolidated Framework for Implementation Research (CFIR) [9] using a directed content analysis approach. A directed content analysis approach is helpful as it allows the researchers to use existing taxonomies to code data, and assists in making that data more translatable to other contexts, thus enhancing the generalizability of the results. Consensus discussions were held to resolve disagreements, develop a shared understanding and to refine the coding. Following consensus discussions and the finalisation of coding, data were extracted into an Excel spreadsheet and researchers independently evaluated each coded piece of text to decide if it were a barrier or enabler. At this time, each researcher also inductively summarised each text artefact into a statement that captured its meaning. Using both a deductive and inductive approach at different stages facilitated analysis that allowed for the CFIR-ERIC framework mapping, and reporting of how the constructs were reflected in the study setting. For example, when participants were giving a response that demonstrated barriers to accessing appropriate training and resources, it was allocated the appropriate summary theme to provide context. A third researcher (JP) assisted with reviewing and refining the coded themes. After the finalisation of barrier and enabler statements, a prioritisation exercise was undertaken to provide a focus for enhancing implementation. Barriers to be addressed and enablers to be amplified were prioritised according to the cumulative majority and selected for mapping to the ERIC tool [11]. The ERIC tool was used to select theory-informed strategies that should enhance implementation. See Figure 1 for an overview of the analysis process.

## 3. Results

### 3.1. Demographics of Included Participants

Twelve nurses of varying levels of experience (3–25 years) participated in the interviews comprising of four nurse managers and eight clinical nurses (including enrolled nurses (EN), registered nurses (RN) and clinical nurse specialists (CNS). The included participants have worked across a range of medical and surgical nursing contexts. They are summarised in Table 1 below.

### 3.2. CFIR Domains and Enabler Constructs (Directed Content Analysis)

Twenty-four (*n* = 24) constructs from five (*n* = 5) domains were mentioned as enablers across the interview data (see Appendix A). Eight constructs across all five CFIR domains were selected for prioritisation and mapping as they represented the cumulative majority of interview participants. Table 2 summarises the enabler coding results.

### 3.3. CFIR Domains and Barrier Constructs (Directed Content Analysis)

Twenty-seven (*n* = 27) constructs from five (*n* = 5) domains were mentioned as barriers across the interview data. Seven (*n* = 7) constructs from four (*n* = 4) CFIR domains were selected for prioritisation and mapping. Table 3 summarises the barrier coding results.

### 3.4. ERIC Strategy Mapping

CFIR-coded barriers and enablers were mapped to the ERIC tool, enabling strategy selection. The ERIC tool is based on expert consensus, and there is a wide variation in agreement as to which strategies best address CFIR constructs. In our study, strategies were selected based on the level of agreement that they should be effective in addressing barriers or enhancing enablers. Strategies that were listed as having the highest percentage of agreement were selected. Where strategies were close in number (e.g., 71% and 76% agreement for addressing the barrier of patient needs and resources), more than one was selected. After this analysis, twelve (*n* = 12) strategies emerged as potentially suitable to enhance implementation, which are summarised in Table 4.

After mapping the barriers and enablers to the ERIC tool, they were then clustered as suggested by Waltz et al. [12]. Twelve (*n* = 12) ERIC strategies were able to be clustered into six (*n* = 6) focus areas, which are summarised in Table 5.

## 4. Discussion

This study used the CFIR-ERIC mapping process to develop an implementation enhancement plan. To the authors’ knowledge, it is the first paper to not only consider the barriers to implementation but also the enablers. Including the enablers in the analysis provides an opportunity to both address issues impeding optimal implementation and enhance the current approach. For example, while education has been provided, the way in which it is delivered may need to be changed to increase the uptake of the intervention. Common barriers and enablers were identified across participant responses. The ERIC tool gave useful guidance to develop a plain language and focused implementation enhancement plan.

The CFIR-ERIC mapping process was selected for this study as it builds on previous work undertaken by a member of the research team [14]. Delaforce and colleagues developed an implementation plan to improve anaemia screening uptake and then tested the utility of the plan in a large hospital setting [15]. Across six months, they significantly improved the odds of receiving appropriate care 10-fold (odds ratio 10.6 [95% CI: 4.406, 25.496] *p* < 0.000)). This result shows that using the CFIR-ERIC approach can have a strong impact on implementation success. Our study extends this approach by also considering the enablers to implementation.

### 4.1. Enablers

Both barriers and enablers were factored into the implementation enhancement plan. Some enablers, including access to knowledge and information, as well as patient needs and resources were also considered barriers for various reasons. Constructs that were coded as enablers only and selected for prioritisation included leadership engagement, cosmopolitanism, relative advantage, formally appointed internal implementation leaders, knowledge and beliefs about the intervention and self-efficacy.

Access to knowledge and information concerns stakeholder access to digestible information about an intervention and how it can be incorporated into workflows [9]. While some staff lamented that there were insufficient training and resources, there was also some mention that support was available from the digital fall prevention workflow vendor and that there were accessible resources both paper-based in the form of a manual and cheat sheet, as well as digital resources on the hospital intranet that staff could reference (See Appendix A). Drawing on ERIC, the strategy to address this is to conduct educational meetings [2]. Using formal education sessions to help educate staff has been shown as effective in changing the motivation levels of staff to use fall prevention interventions [16]. These could be supplemented by the provision of easily accessible and digestible information to support adoption during the early implementation of an intervention [17]. These resources should be developed in consultation with the end users to maximise the chance of adoption. Neglecting to sufficiently engage with staff during the development of educational material has been demonstrated to contribute to the failed implementation of falls education [18].

Patient needs and resources considers the extent to which patient needs, as well as barriers and enablers to meet those needs, are accurately known and prioritized by an organization [9]. Some staff were concerned that they could not always meet patient needs when multiple people were buzzing at the same time as well as when the technology was misused by cognitively impaired patients. However, they also said that they felt they could provide improved care due to the ability to answer patient concerns quickly and provide reassurance. Within healthcare, and particularly for the nursing population, staff must see the benefit of an intervention for their patients to ensure uptake [19]. The provision of information and reports that show the improved care that results from the fall prevention platform will help ensure ongoing engagement [9,19]. In addition to sharing the improvements in patient care, the ERIC tool suggests that obtaining patient feedback and involving them during implementation may also assist in increasing the effectiveness of implementation attempts [2]. Previous studies engaged patients to help design fall prevention toolkits and achieved a 34% reduction in injurious falls [20,21].

Leadership engagement is crucial for any organisational change. In this study, leadership engagement was an enabler as several executive champions had a vested interest in seeing an improvement in fall prevention. A descriptive study that surveyed 60 hospitals in the United States found that 100% of respondents advised that they used at least one strategy to engage leadership to set expectations [6]. Strategies included updating fall policy and procedure, including falls in annual reports and embedding fall reporting into key roles such as the hospital safety direction [6]. This is consistent with the ERIC tool and suggests involving executive boards as well as identifying and preparing champions [2].

Cosmopolitanism as a construct concerns itself with the degree to which a facility is networked externally [9]. Cosmopolitanism emerged as an enabler as staff, having also worked in other hospitals within the region, highlighted that they are the first in the region to use the technology. Greenhalgh and colleagues, in their seminal systematic review, note that cosmopolitanism can have downstream effects when other hospitals see the success of an innovation and are more likely to adopt it [17]. The encouragement of external “boundary spanning” roles among staff should be promoted to help facilitate awareness of how other facilities operate and ensure the dissemination of shared learnings [17]. The study setting is one of numerous health facilities in the district and was chosen as the trial site. Progress reports are routinely shared with other district hospitals, which is consistent with the ERIC tool suggestion to build a coalition through the recruitment and cultivation of relationships with partners in the implementation effort [2].

Relative advantage is the degree to which stakeholders perceive an innovation as being more beneficial than an alternative solution [9]. Relative advantage emerged as an enabler where participants explained how they felt patient care, communication and accountability were all improved when using the Concentric Care fall prevention platform. In a Canadian study that analysed factors influencing the uptake and sustainability of best practice guidelines (including fall prevention), the degree to which nurses saw the advantages of using the guidelines was strongly related with ongoing uptake of best practice guidelines [22]. Sense checking with staff before implementation and the provision of information on outcomes may help enhance the level to which an intervention is perceived to have a relative advantage. The ERIC tool also suggests identifying and preparing champions to enhance this aspect of implementation [2]. Health facilities should consider identifying champions and ensure that part of their role includes the delivery of positive patient outcomes.

Formally appointed internal implementation leaders includes organisations formally identifying and appointing staff to be responsible for the implementation of an intervention [9]. Staff reported that the nurse unit managers had responsibility for ensuring the digital fall prevention workflow was being used in daily practice. Ensuring that first-level leaders demonstrate implementation behaviours that encourage the adoption of an intervention has been shown to improve uptake [23]. One study used an interrupted time series to measure the effect of a fall champion assigned to perform audits within a unit of a large hospital and found a significant reduction in falls following the commencement of the program (3.67 falls per 1000 patient days pre, 1.36 post) [24]. In alignment with the ERIC tool [2], ensuring that either clinician champions or first-level managers are identified and supported to encourage implementation is necessary to maximise the chance of practice uptake.

Knowledge and beliefs about the intervention is the target users’ attitude towards and value placed on the intervention as well as familiarity with information related to the intervention. Multiple staff acknowledged that they feel the system has made an improved difference to patient care. To reinforce this belief, it will be important for the health facility to ensure information on patient outcomes (including rate of falls, injury, length of stay, etc.) is fed back on a continuing basis to help maintain engagement. A study of 162 nurses from South Korean hospitals found a positive correlation between nurses’ attitudes regarding patient falls and the execution of fall prevention activities [25]. As suggested by the ERIC tool [2], there is a need for consistent, repeated and customised fall prevention education to promote engagement in fall prevention activities.

Self-efficacy is the belief of target intervention users in their own capabilities to execute courses of action to achieve implementation goals [9]. Staff reported that the new environment where the fall prevention platform has been implemented is different to previous ward layouts as there are single and double rooms, as opposed to four bed “hob” rooms. Staff reported that without the system, they do not feel that they could provide the same standard of safe care in the new configuration. Aspects of the system act as a reminder system or “safety net” for the staff to ensure they are monitoring patients at high risk of fall closely. Given that there is low-quality and uncertain evidence to suggest that alarms alone contribute to a reduction in falls, it is possible that the multifaceted aspects of this system, including the ability to talk to patients in real time to offer reassurance, may act as an effective reminder to staff [4]. The ERIC tool suggests that providing ongoing consultation and training that is dynamic can positively influence the implementation of new interventions [2]. This is supported by a stepped-wedge, cluster-randomized controlled trial run over a 50-week period, including 3606 patient episodes [26]. The study used highly individualized staff and patient education to almost halve the number of falls (pre, *n* = 196, 7.8/1000 patient days vs post *n* = 380, 13.78/1000 patient days. Adjusted rate ratio 0.60 [95% CI 0.42–0.94], *p* = 0.003) [26]. Ongoing training should be adopted by health facilities wishing to implement new interventions.

### 4.2. Barriers

The analysis revealed seven key barriers to implementation including: access to knowledge and information, available resources, compatibility, patient needs and resources, design quality and packaging, adaptability and executing [9]. Access to knowledge and information was mentioned as a barrier in the context of not getting to practice using the system or engage in face-to-face training. A 2020 systematic review assessed the effectiveness of digital education on improving nurses’ knowledge in relation to chronic wound management and found that a blended approach that used both digital and in-person modalities was more superior [27]. As the world emerges from the COVID-19 pandemic, facilities should prioritise a blended learning approach to ensure nursing staff are receiving quality education that supports the provision of safe, evidence-based care. This approach is congruent with the ERIC tool [2] suggestion to conduct educational meetings [2]. Future testing of this approach will confirm its level of utility.

Available resources was a barrier in the context of reduced staffing because of the ongoing COVID impacts and the desire for more equipment. As per the seminal work of Michie et al., in order for a clinical innovation to be used, there needs to be sufficient capability, opportunity and motivation [28]. In this context, increased patient to nurse ratios, the result of staff furloughs and workforce shortages rendered staff incapable of adequately engaging with the platform at all times due to an overwhelming workload. This problem has been felt worldwide as health services struggle through the increased demand and reduced workforce resourcing due to the ongoing COVID-19 pandemic. A recent qualitative study that interviewed 295 nurses from the Central Region of the Philippines to understand reasons for missed care confirmed that adequate staffing levels are crucial to providing safe care [29]. Until the impacts of the COVID-19 pandemic subside, it will be difficult to address this issue. Participants also stated a desire for additional equipment. Staff generally saw the benefit of the Concentric Care fall prevention platform but commented it was sometimes hard to obtain a bed equipped with the technology. This is consistent with previous research noting insufficient resources are a key barrier to the use of hospital-based interventions [30]. As suggested by the ERIC tool [2], investment into additional resources such as additional equipment is worthy of consideration to enhance the provision of fall prevention beds.

Compatibility was reported as an issue where staff felt that they needed to develop new workflows to adapt to the new environment, inclusive of the Concentric Care fall prevention platform. This is consistent with previous research findings that when workflows are not sufficiently developed and communicated to staff, particularly in the context of digital health interventions, their uptake and use can be impeded [31]. The ERIC tool [2] suggests promoting adaptability to enhance uptake. One way of applying this strategy is to perform a clinical workflow analysis to support the implementation of eHealth interventions including identifying discrete workflow components, workflow assessment, triangulation and stakeholder proposal of intervention implementation [31]. Facilities intending to implement the Concentric Care fall prevention platform should undertake this mapping exercise beforehand and ensure all staff are trained to see how they need to adapt their work practices.

Patient needs and resources was listed as a barrier in the context of both the complexity of the handset and the impact of additional workloads on nurses’ ability to provide safe care. Nurse participants stated that often patients seemed to misuse the handsets; in particular, those who are cognitively impaired. A 2013 stepped-wedge, cluster-randomized controlled trial tested the impact of an individualised education program on fall prevention and included patients who were cognitively intact or showed a mild impairment [26]. The study, which included 3606 participants, found that individualised patient education programmes combined with training and feedback to staff reduced the rate of falls by almost half (*n* =196, 7.80/1000 patient days vs. *n* = 380, 13.78/1000 patient days) [26]. The ERIC tool [2] suggests that obtaining patient feedback and involving them in implementation can address this barrier. Consideration should be given to revising current patient education as well as staff training and feedback mechanisms as it may have a positive impact on proper system use, as well as generally helping patients to be more aware of strategies to reduce their risk of fall [32].

Design quality and packaging was reported as a barrier in the context of noise fatigue, and the difficulty in cancelling alarms/integrating the new system into existing workflows. Nurses lamented the fatigue they felt when their phones would constantly ring and difficulty with deactivating bed alarms. Noise fatigue is not a new problem and has been reported in the contact of a myriad of interventions (call bell, telemetry, vital sign monitoring) [33,34]. Various approaches can be used to overcome alarm fatigue including modification of alarm parameters or clinical workflow re-design with some studies showing a 68% improvement in alarm notifications after using improvement cycles to re-design workflows with multidisciplinary input [35]. The ERIC tool [2] suggests promoting adaptability to address this barrier and it is recommended that facilities work closely with vendors to tailor solutions to local needs.

Adaptability was raised as a barrier in the context of integrating the system into existing workflows and ensuring contingencies were in place for when the system is down. Both issues are interconnected, and sufficient planning and education is needed so that end users know exactly how the system can complement existing workflows, and what needs to happen when the system is down. As the integration of digital interventions into healthcare workflows increases, there will be an ongoing need to ensure that formal contingency plans are in place when technologies fail and that all stakeholders are aware of their existence [36]. The ERIC tool [2] suggests promoting adaptability and facilities could consider integrating downtime simulation into existing education plans to ensure that staff can provide safe and quality care, even when digital systems are down. Reporting of the methodology for, and the effectiveness of contingency training remains limited, and should be a focus for future research [37].

Executing as a construct considers the degree to which an intervention is used as intended and whether implementation occurs according to plan [9]. Staff reported that it was hard to fully engage with the platform when managing competing demands and sometimes prioritise other cares or tasks above using the system as it is intended. While recognising that there will inevitably be times where staff cannot fully engage with a system as intended, opportunities to reduce this as a mediating barrier to use can be operationalised using a team learning approach. A seminal study in 2001 that reported factors of success when implementing technology into hospitals, examined 16 top performing hospital cardiac units and the degree to which implementation of a new technology had been achieved as was intended [38]. The study developed a model based on factors that created an environment for success including four key areas: deliberate selection of team members for implementation with overt discussion about who should be involved and why; undertaking “dry runs” making sure to discuss the technology and communication; implementing new forms of team communication; and using active team discussion on how the new technology intervention was going [38]. These four areas can be operationalised by using collective, team learning opportunities and should be considered when implementing this or other similar digital interventions that will require a change in the way teams communicate and work with each other [38]. Such strategies are supported by the ERIC tool [2,39], which proposes that a purposeful re-examination of implementation can help address the barrier of executing an intervention. The enhancement plan generated by this study will guide changes in the implementation of the Concentric Care fall prevention platform.

### 4.3. Limitations

The main limitations of this study are the method used and sample size. Qualitative data are rarely generalisable to other contexts. However, it is becoming increasingly apparent within the discipline of implementation science that context must always be considered, and variability embraced. Within this study, this limitation is partly offset by the use of the CFIR-ERIC tool, which provides a taxonomy that can be readily applied to other contexts wishing to implement the Concentric Care fall prevention platform (or similar). In terms of the sample size, only 12 participants were recruited and therefore the data may not be representative of all staff involved with the implementation or daily use of the Concentric Care fall prevention platform.

### 4.4. Future Research

These results will be shared with the study site, where a workshop to agree on what can and cannot be adopted to improve uptake and engagement with the Concentric Care fall prevention platform will occur. Following the workshop, future evaluation using a type-two hybrid effectiveness design to measure the impact of this process will be reported once complete.

## 5. Conclusions

The CFIR-ERIC approach provided clear and concise recommendations to enhance implementation, which are also supported by the evidence. Given the close agreement between theory and evidence, it is feasible that the application of this implementation enhancement plan is likely to result in improved uptake and engagement with the Concentric Care fall prevention platform. The results of this study will provide a blueprint to enhance implementation that will be tested for effectiveness at a later stage.

## Figures and Tables

**Figure 1 ijerph-20-03794-f001:**
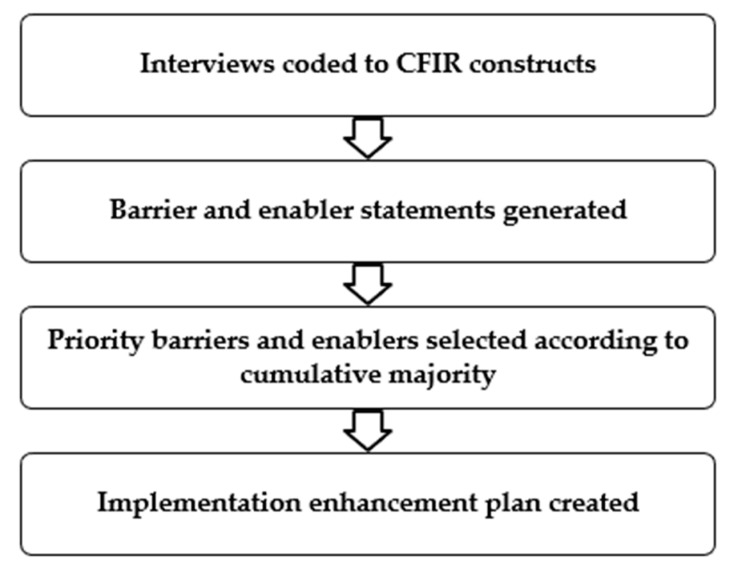
Data coding and barrier/enabler mapping process.

**Table 1 ijerph-20-03794-t001:** Demographics of participants.

		N
Role	Nurses (EN/RN/CNS)	8
	Nurse Manager	4
Gender	Female	10
	Male	2
Ward	Surgical	5
	Medical	6
	Other (N/A)	1

Note: Average years of experience 9.25 years.

**Table 2 ijerph-20-03794-t002:** Enabler coding results.

Consolidated Framework for Implementation Research Enabler Domain	Construct (*n* = Participants)	Enabler Statement Theme	Exemplar Quote (Role, Years of Experience)
Intervention characteristics	Relative advantage (*n* = 12)	Utility of information provided by system System makes processes more efficient Perceived and real quality of care improved Team communication and accountability enhanced	“I also have access to like the back sort of section of the responder system so I can see what calls are made in within an hour or weekly period and I can see the peak periods and things like that so it’s interesting, across different wards the calls and the need for nurses are different during different times” (Nurse manager, 12) “I think they’re reassured knowing that I’ve answered their buzzer within like under 2 min which is quite good. I think most of our response time is like 30 s which is yeah so like we could just be like and tell them what we’re doing, not saying that we’ve attended to their needs in 30 s but we’re saying this is what we’re doing, we’re going to do this next and we’ll get your pain relief or something like that later. I think it’s good” (Nurse, 13) “I find it’s been very handy for the nurse in regards to contact the doctors… I feel like we’ve definitely lifted the pressure on the in charge, because we’re doing a lot of those important phone calls ourselves because we have it at hand. Instead of passing that task along.” (Nurse, 3)
Inner setting	Access to knowledge and information (*n* = 11)	Appropriate and accessible training and support Formal communication channels	“Yeah, there’s been support, I think enough support. And they’re still getting the educators for the new staff and new graduate students to go through the Responder system and everything. And that’s included in their orientation pack” (Nurse manager, 10) “… reinforcing it in safety huddles maybe and orientation for new staff…” (Nurse, 3)
	Leadership engagement (*n* = 9)	Leadership is engaged in the implementation, monitoring and evaluation of the system.	“From a hospital leadership point of view it is seen as very important. From a district wide executive leadership level, it is seen as very important, and they’re very interested. They have done some rounding, because they want to know how it works, they have a vested interest to roll out across the district. They have some concerns in terms of how well we’re integrating it, but definitely a high level of interest.” (Nurse manager, 3)
Outer setting	Patient needs and resources (*n* = 8)	Perceived and real quality of care improved Orientating patients to the system is key	“Yeah it’s definitely easier to challenge their expectations, because you can say okay it’s just as simple as answering the phone saying okay I’ve just got to do this, I’ll be with you in 5 min. Like it’s a lot easier for them I think.”(Nurse, 3) “I think so because when a patient gets admitted to the ward, you introduce yourself and you introduce the patient’s bed and surroundings, and also the buzzer, and you make sure that they know switch to, because the buzzer says if you want nurse for pain or water or toileting you can press that. So it’s sort of clear in the buzzer itself, so if they are alert and oriented and they’re able to use that.” (Nurse manager, 10)
	Cosmopolitanism (*n* = 5)	Awareness of the technology being new, cutting edge or providing an advantage to the health facility	“No I think that we’re one of the first within our region. I don’t know if it’s our region or our LHD to implement this so I guess therefore we’re the test subjects…” (Nurse manager, 12)
Characteristics of individuals	Knowledge and beliefs about the intervention (*n* = 5)	Perceived and real quality of care improved Team communication and accountability enhanced System is perceived as easy to use	“I think it’s a good system and it’s, if we look at your falls numbers you can see a real difference from the old hospital to the new hospital—so in that respect it’s quite good, and also if you look at the noises that are on it like you used to have patient buzzers and now it’s coming into a phone, so it helps with that noise pollution also. So, I think it’s a good system and we all have time where we needed to adapt to that, and I think we’re sort of comfortable like even allocating to the staff and yeah I think it’s working.” (Nurse manager, 10)
	Self-efficacy (*n* = 5)	Staff feel more empowered to do their job efficiently Staff feel confident to use the system and help	“Well when we moved to the new hospital the way the ward was set out, we completely eliminated the availability of hob (4 bed) rooms, so I feel like if we didn’t have the Rauland system it would be impossible. Because we were able to set it up so you don’t set and, not forget, but like just set and go and they’re quite sensitive that we can rely that the alarms will sound when they need to if they’ve been set up correctly.” (Nurse, 3)
Process	Formally appointed internal implementation leaders (*n* = 5)	Leadership champions Internal working parties including leaders help support uptake Leadership are engaged in the implementation, monitoring and evaluation of the system	“…our NUMS are the main role models for us and they’re the ones that are implementing systems like this and ensuring it’s happening safely and I think the team leaders and our senior clinical staff like we need to just ensure that they’re all sort of like giving up and educating everyone consistently and staying up to date with everything if there’s any changes and stuff like that”. (Nurse, 13) “So team leaders, ward staff that know how to work it and the NUMs basically. They walk through and push people to use it properly.” (Nurse, 3)

**Table 3 ijerph-20-03794-t003:** Barrier coding results.

CFIR Barrier Domain	Construct (*n* = Participants)	Barrier Statement Theme	Exemplar Quote (Role, Years of Experience)
Inner setting	Access to knowledge and information (*n* = 11)	Lack of appropriate and accessible training and support/resources	“I moved on the first day… it was kind of teach as you go, I found…I don’t know if I’ve just missed something or if I’ve missed a training day, but I only recently was shown how to use the falls alert, like how to properly do that. I don’t know if that was just me missing something.” (Nurse, 3)
	Available resources (*n* = 8)	COVID impacts on staffing and change management Lack of appropriate and accessible training and support/resources Operational resources	“We have had an increase in falls recently, but I don’t think that’s anything to do with the Responder system I think that’s more to do with staffing levels.” (Nurse, 7) “There’s never enough Wi-Fi beds for the patients.” (Nurse, 9)
	Compatibility (*n* = 8)	COVID impacts on staffing and change management Design and function of the buzzer Lack of consultation/piloting New ways of working as a team are needed Workflow creates excess noise, waste and infection control risks	“COVID interruptions and the inability to saturate normally in a change management process you would have you know saturation, multiple touch points, we weren’t able to do that it was a one stop shop or you went onto the Rauland’s website and did your training virtually.”(Nurse manager, 25) “The old system where we sat you could see up and down your corridors and you could see your buzzers. So, it’s not just about the Rauland itself, it’s also about the way the hospital itself is set up to fit in with these systems that are getting put in place for us to use.” (Nurse, 14)
Outer setting	Patient needs and resources (*n* = 8)	Design and function of the buzzer Increased nurse workload due to technology Operational resources Workflow creates excess noise, waste and infection control risks	“…it can be quite overwhelming if you have a couple…of unwell patients that you’re dealing with and are constantly calling and you have so many tasks, your list is lining up and they keep continuing to call. It can get very overwhelming.” (Nurse, 3) “With our cognitively impaired patients, patients that are hard of hearing, they don’t know whether they’re pressing, they’re pressing it and it’s constantly pressing, and it’s constantly alarming…” (Nurse, 9)
Intervention characteristics	Design quality and packaging (*n* = 10)	Design and function of the buzzer Lack of consultation/piloting Lack of appropriate and accessible training and support/resources New ways of working as a team are needed Operational resources Workflow creates excess noise, waste and infection control risks Contingencies when system down need to be developed	“Not so much the phones but definitely the alarms, the amount of noise that it makes. Something else with the phones too is that there’s just so much noise. You know you’re getting tired of the bleeping; you’re getting tired of the you kind of get worn out listening to it and then you start ignoring it.” (Nurse, 3) “I don’t love the system. I find it can be quite difficult to use at times. I find the reception is not ideal and I find there’s a few things that could definitely be improved to make it more user friendly.” (Nurse, 9) “… I asked … to order some chair alarms because they can’t be in bed all the time.” (Nurse manager, 10)
	Adaptability (*n* = 7)	Contingencies when system down need to be developed New ways of working as a team are needed Workflow creates excess noise, waste and infection control risks	“Where it’s at at the moment is really trying to integrate it fully into our care. In terms of the use of the system and the phones, that’s definitely a day in day out use, but it’s incorporating the beds and the phones and setting it up appropriately for the high falls risk patients. While we’re using it I don’t think it’s fully integrated yet, and the model of care that’s been put together around it, from a local model of care, we haven’t fully communicated that, and that has contributed to it not being fully integrated yet. In terms of how we use it to the best of our ability and how we bring it, the system and the equipment into patient care and you know into our team leader role, and into our hourly rounding, how it’s fully integrated and incorporated into the other cares that we do.” (Nurse manager, 3) “Oh, right yeah. It does worry me and the more systems we put online—I do get nervous around what that might look like and I do know that you know if the call bell system goes down that the phones at the desk can be put in place and we’ve got a business continuity plan where we have to get an extra nurse that sits at the desk and uses the phone system to be able to do all the call bells so I do know there’s a business continuity plan but that’s probably a bit clumsy to be honest in a disaster, so I don’t think it’s that solid but that is a business continuity plan.” (Nurse Manager, 25)
Process	Executing (*n* = 7)	COVID impacts on staffing and change management Contingencies when system down need to be developed Increased nurse workload due to technology Operational resources Lack of appropriate and accessible training and support/resources Lack of consultation/piloting New ways of working as a team are needed Workflow creates excess noise, waste and infection control risks	“On a good shift everything flows, as I said when you have a tricky balance where you’ve got very unwell patients, when you’ve got rapid responses, when you’ve got arrivals and departures happening all at once… it can be quite stressful, it can be quite chaotic and buzzers going left right and centre…and I guess when the phones escalate and it turns, and the buzzers end up extending that warning buzzer noise where everyone has to drop everything and work out okay who’s doing what where, where is this coming from, it can—and it’s also because staff at the time might be in the middle of fetching medications down from pharmacy, they might be helping, assisting a patient safely to you know leave once they’ve been discharged from hospital, and also settling the new patients in and you know sourcing relevant things like personal protective equipment if it’s an infectious patient and little things like that. So all them (sic) factors basically can also add to the delay….” (Nurse, 9)

**Table 4 ijerph-20-03794-t004:** Summary of enabler and barrier constructs mapped to ERIC strategies.

CFIR Construct (Prioritised)	Barrier	Enabler	ERIC Strategy (Most Strongly Recommended)	% Agreement
Available resources	**√**		Access new funding	78%
Compatibility	**√**		Promote adaptability	45%
Design quality and packaging	**√**		Promote adaptability	48%
Adaptability	**√**		Promote adaptability	73%
Executing	**√**		Purposely re-examine the implementation	45%
Access to knowledge and information	**√**	**√**	Conduct educational meetings	79%
Patient needs and resources	**√**	**√**	Obtain and use patients/consumers and family feedback Involve patients/consumers and family members	76% 71%
Leadership engagement		**√**	Involve executive boards Identify and prepare champions	45% 41%
Cosmopolitanism		**√**	Build a coalition	62%
Relative advantage		**√**	Identify and prepare champions	45%
Formally appointed internal implementation leaders		**√**	Identify and prepare champions	64%
Knowledge and beliefs about the intervention		**√**	Conduct educational meetings	56%
Self-efficacy		**√**	Provide ongoing consultation Conduct ongoing training Make training dynamic	41% 41% 41%

**Table 5 ijerph-20-03794-t005:** Implementation enhancement plan.

ERIC Cluster [12]	ERIC Strategy [11]	Definition [11]
**Train and educate stakeholders**	Conduct educational meetings	Hold meetings targeted toward different stakeholder groups (e.g., providers, administrators, other organizational stakeholders, and community, patient/consumer and family stakeholders) to teach them about the clinical innovation
	Provide ongoing consultation	Provide ongoing consultation with one or more experts in the strategies used to support implementing the innovation
	Conduct ongoing training	Plan for and conduct training in the clinical innovation in an ongoing way
	Make training dynamic	Vary the information delivery methods to cater to different learning styles and work contexts, and shape the training in the innovation to be interactive
**Utilize financial strategies**	Access new funding	Access new or existing money to facilitate the implementation
**Adapt and tailor to context**	Promote adaptability	Identify the ways a clinical innovation can be tailored to meet local needs and clarify which elements of the innovation must be maintained to preserve fidelity
**Engage consumers**	Obtain and use patients/consumers and family feedback	Develop strategies to increase patient/consumer and family feedback on the implementation effort
	Involve patients/consumers and family members	Engage or include patients/consumers and families in the implementation effort
**Use evaluative and iterative strategies**	Purposely re-examine the implementation	Monitor progress and adjust clinical practices and implementation strategies to continuously improve the quality of care
**Develop stakeholder interrelations**	Identify and prepare champions	Identify and prepare individuals who dedicate themselves to supporting, marketing and driving through an implementation, overcoming indifference or resistance that the intervention may provoke in an organization
	Build a coalition	Recruit and cultivate relationships with partners in the implementation effort
	Involve executive boards	Involve existing governing structures (e.g., boards of directors, medical staff boards of governance) in the implementation effort, including the review of data on implementation processes

## Data Availability

The datasets during and/or analysed during the current study are available from the corresponding author on reasonable request.

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
