# Peer review of "Creating an Implementation Enhancement Plan for a Digital Patient Fall Prevention Platform Using the CFIR-ERIC Approach: A Qualitative Study"

_ijerph, 2023, doi:10.3390/ijerph20053794_

Round 1
Reviewer 1 Report
I was very excited to read this novel study. There are several areas which need improvement but overall it is a good contribution to the literature if these areas are addressed. See below.
INTRODUCTION
lines 58-59 - suggest removing bed alarms as a prevention strategy as research is inconclusive. Also suggest remove low beds as that is an injury prevention strategy not a fall prevention strategy.
lines 90-99 - better references are recommended. refer to eric website
-
Powell et al 2015: This article lists all 73 ERIC strategies with short descriptions. Longer rationale and descriptions are documented in Additional File 6 published with this article.
-
Waltz et al 2015: This article groups 73 ERIC strategies into 9 clusters based on concept mapping methods.
-
Perry et al 2019: This article lists recommended refinements to the list of ERIC strategies based on findings from the ESCALATES National Evaluation.
METHODS
-Clinical innovation - please provide a reference for the technology platform. -please indicate how it interfaces with EMRs.
-more information needs to be described on this clinical innovation to provide the readers context for tables 1 and 2. this could be done in-text or in a supplementary document. for example, if you were describing this to someone who knows nothing about it, how would you describe it and where could the reader also look up more information on it?
-please add if the training was REQUIRED by staff or not.
-move IRB info up to beginning of participants and recruitment section
-data collection - please describe the types of questions asked in the interview and attached the guide as a supplementary document
DATA ANALYSIS
-Demographics - please include average and age range, sex, and race/ethnicity table of participants. Also, each quote should be attributed to a participant number so that the reader can see if there were differences based on participant demographics. (if this is possible)
Table 1
-use brackets to spell out any acronyms for the reader
Mapping - it is unclear why some constructs and domains were chosen to map and others were not. please include this info somewhere
Font should be the same in all of the tables and readabiltiy would be better if items were left justified.
GENERAL COMMENT
-which enabler-matched strategies were actually used by the health system in the implementation process? Consider adding an * and footnote to indicate this for table 3 and include this information in the discussion how the enabler strategies mapped by eric post implementation were different from the actual enablers with real implementation..
DISCUSSION
In general, I find the discussion lacking with reference to other literature on inpatient falls prevention. Please revise.
line 246 - provide reference for vendor
line 273 - reference
lines 283-284 - you use encourage twice. reword
lines 287-297 - are there any fall prev inpatient studies you can cite about relative advantage?
same for the next paragraph.
line 345- resources should not be caps or decide if you are going to caps the constructs or not in the discussion as it is not consistent throughout.
ERIC tool
this section is greatly lacking. The authors need to discuss the strategies that were mapped to the barriers and facilitators and if they have been used effecitvely in prior falls prevention research or not.
More discussion needs to be
Author Response
Dear Reviewers,
Thank you very much for your considered feedback. Your time in providing this is greatly appreciated and your suggestions have improved the quality of the manuscript.
During review, it was acknowledged that two additional authors should be added to the manuscript given their contributions to the conceptualisation of the project. We also have moved the order of authors as Associate Professor Joy Parkinson provided assistance with conceptualisation of the discussion as it is now.
Please see below our response to reviewer comments:
Reviewer One:
I was very excited to read this novel study. There are several areas which need improvement but overall, it is a good contribution to the literature if these areas are addressed. See below.
Thank you for your kind comments.
INTRODUCTION
lines 58-59 - suggest removing bed alarms as a prevention strategy as research is inconclusive. Also suggest remove low beds as that is an injury prevention strategy not a fall prevention strategy.
Thank you for your suggestion, we have removed the low beds but left the environmental or assistive technologies as there is an explanation qualifying the quality of evidence currently available – which, as you rightly state, is low across the board.
lines 90-99 - better references are recommended. refer to eric website
- Powell et al 2015: This article lists all 73 ERIC strategies with short descriptions. Longer rationale and descriptions are documented in Additional File 6 published with this article.
- Waltz et al 2015: This article groups 73 ERIC strategies into 9 clusters based on concept mapping methods.
- Perry et al 2019: This article lists recommended refinements to the list of ERIC strategies based on findings from the ESCALATES National Evaluation.
Thank you – we have made changes so that it is clearer as to what each statement is referring to and the website is now specifically referenced.
METHODS
-Clinical innovation - please provide a reference for the technology platform. -please indicate how it interfaces with EMRs.
-more information needs to be described on this clinical innovation to provide the readers context for tables 1 and 2. this could be done in-text or in a supplementary document. for example, if you were describing this to someone who knows nothing about it, how would you describe it and where could the reader also look up more information on it
Thank you, we have now included a supplementary file that details the features.
-please add if the training was REQUIRED by staff or not.
We have now clarified that staff training was mandatory at line 137.
-move IRB info up to beginning of participants and recruitment section
We have now moved this up to the beginning at line 143.
-data collection - please describe the types of questions asked in the interview and attached the guide as a supplementary document
We have now provided a supplementary file with the interview guide.
DATA ANALYSIS
-Demographics - please include average and age range, sex, and race/ethnicity table of participants. Also, each quote should be attributed to a participant number so that the reader can see if there were differences based on participant demographics. (if this is possible)
We have now included more demographic information whilst also preserving the anonymity of participants. There are also role and years of experience assigned to quotes.
Table 1
-use brackets to spell out any acronyms for the reader
We have spelled this out in full now.
Mapping - it is unclear why some constructs and domains were chosen to map and others were not. please include this info somewhere
We include a comment in the data analysis section about this at line 194 “After the finalisation of barrier and enabler statements, a prioritisation exercise was undertaken to provide a focus for enhancing implementation. Barriers to be addressed and enablers to be amplified were prioritised according to the cumulative majority and selected for mapping to the ERIC tool9”
Font should be the same in all of the tables and readability would be better if items were left justified.
Thank you for picking this up – yes, the tables do look much clearer when left justified.
GENERAL COMMENT
-which enabler-matched strategies were actually used by the health system in the implementation process? Consider adding an * and footnote to indicate this for table 3 and include this information in the discussion how the enabler strategies mapped by eric post implementation were different from the actual enablers with real implementation..
Thank you for this excellent question. This study will inform a larger project which will test the implementation enhancement plan devised from this work. In this case, an implementation science informed approach was not used in the beginning, rather, we have now come up with something that should enhance the uptake of the intervention. It would be our preference to save this discussion for the next paper which will use a type-two hybrid-effectiveness design to test the effect and implementation outcomes. It will be much more valuable to comment on the “before and after” here. We have added a statement in the beginning to help signpost this intention to the reader.
DISCUSSION
In general, I find the discussion lacking with reference to other literature on inpatient falls prevention. Please revise.
Thank you, more fall specific literature has been added into the discussion – example, line 345, line 384, line 402, line 435,
line 246 - provide reference for vendor
Thank you – we have now directed readers to supplementary file one
line 273 – reference
Thank you – added.
lines 283-284 - you use encourage twice. Reword
Thank you – addressed.
lines 287-297 - are there any fall prev inpatient studies you can cite about relative advantage?
same for the next paragraph.
Thank you – More fall-specific literature is included here.
line 345- resources should not be caps or decide if you are going to caps the constructs or not in the discussion as it is not consistent throughout.
Thank you – addressed.
ERIC tool
this section is greatly lacking. The authors need to discuss the strategies that were mapped to the barriers and facilitators and if they have been used effectively in prior falls prevention research or not.
Thank you for pointing this out – we have now integrated a comment in each section of the discussion to overtly compare the suggested enablers and how the evidence has overcome them.

Reviewer 2 Report
This study took a qualitative approach using focus groups/interviews (of 12 participants) across 4 inpatient care units to identify barriers and enablers. The qualitative data was used to map to the Expert Recommendations for Implementing Change (ERIC) tool to develop an implementation enhancement plan. This study report was well-organized and clear.
The authors should highlight the areas/barriers and enablers that were not covered in the ERIC tool.
Thank you for the opportunity to review this manuscript.
Author Response
Dear Reviewers,
Thank you very much for your considered feedback. Your time in providing this is greatly appreciated and your suggestions have improved the quality of the manuscript.
Reviewer two comments:
This study took a qualitative approach using focus groups/interviews (of 12 participants) across 4 inpatient care units to identify barriers and enablers. The qualitative data was used to map to the Expert Recommendations for Implementing Change (ERIC) tool to develop an implementation enhancement plan. This study report was well-organized and clear.
 
The authors should highlight the areas/barriers and enablers that were not covered in the ERIC tool.
 
Thank you for the opportunity to review this manuscript.
Thank you for your comments. In the discussion, we have expanded the exploration of what the ERIC tool results were in the context of other ideas suggested in the evidence. We found that suggestions from the evidence matched the ERIC recommendations closely and all were mentioned as being effective. As our approach was to map the barriers to the ERIC framework using the CFIR, and all the CFIR constructs that were have a suggested ERIC strategy, there were none missing and thus, we are unable to elaborate on this.

Reviewer 3 Report
1. The research gap in this study is not fully addressed, and how to bridge the gap is not clear explained in this research. For example, line 55-77 confuse whether the intervention is effective and what intervention is the most effective. At the same time, it expounds that the problems existing in the current research include "On a single site only and deliver interventions", but this is the deficiency in this paper as well.
2. The background in the part of introduction need further sufficient, the references should be updated.
3.The method part should supplement the reasons and advantages of choosing direct content analysis rather than other qualitative analysis method like thematic framework analysis, and supplement the references of methods.
4.It mentioned “To the authors knowledge, it is the first paper to do so considering not only the barriers to implementation, but also the enablers.”(line220), but the advantages of considering enablers so were not elaborated.
5.The contribution of the study are not discussed clearly. It is recommended to clarify the contributions of the research in results part, including case assessment results(Concentric Care fall prevention platform.), method( CFIR-ERIC mapping process) and tools ( CFIR ERIC).
6.The quantitative analysis results of content analysis are not discussed.
7. The content in “Barrier statement theme” of Table 2. Barrier coding results(line201)need to be further refined.
8.It is suggested to explain clearly the contribution of the research results from three levels In the part of discussion: 1) case assessment results; 2)assessment methods; 3) assessment tools.
Author Response
Dear Reviewers,
Thank you very much for your considered feedback. Your time in providing this is greatly appreciated and your suggestions have improved the quality of the manuscript.
Reviewer Three:
- The research gap in this study is not fully addressed, and how to bridge the gap is not clear explained in this research. For example, line 55-77 confuse whether the intervention is effective and what intervention is the most effective. At the same time, it expounds that the problems existing in the current research include "On a single site only and deliver interventions", but this is the deficiency in this paper as well.
Thank you for raising this issue – we have clarified this area further and added some explanation at the end of the introduction that the results of this study will be used to inform a type-two hybrid effectiveness study, which is anticipated to help address the gap of knowing which intervention supported by the chosen implementation strategies are effective.
- The background in the part of introduction need further sufficient, the references should be updated.
We have now added a 2022 systematic review as well as the 2018 Cochrane review.
3.The method part should supplement the reasons and advantages of choosing direct content analysis rather than other qualitative analysis method like thematic framework analysis, and supplement the references of methods.
We have added some clarification around this at line 189.
4.It mentioned “To the authors knowledge, it is the first paper to do so considering not only the barriers to implementation, but also the enablers.” (line220), but the advantages of considering enablers so were not elaborated.
We have added some content to help explain the advantages of this approach at line 294.
5.The contribution of the study are not discussed clearly. It is recommended to clarify the contributions of the research in results part, including case assessment results (Concentric Care fall prevention platform.), method (CFIR-ERIC mapping process) and tools ( CFIR ERIC).
Thank you for your comment – it is intended that the plan devised from this study will be applied in the setting and then measured at a later stage using a type-two hybrid-effectiveness study design. The intent of this paper is simply to outline how we identified enhancements that could be made to improve the implementation of the system. We have added a clarifying sentence to the end of the introduction to help guide the reader to contextualise the study purpose at line 115.
6.The quantitative analysis results of content analysis are not discussed.
Thank you for your suggestion. We have now added a supplementary file that includes a table showing all of the barriers and enablers with the number of participants who mentioned them.
- The content in “Barrier statement theme” of Table 2. Barrier coding results(line201)need to be further refined.
The barrier statement themes are the result of refining and organising by the research team. We are unable to change this further.
8.It is suggested to explain clearly the contribution of the research results from three levels In the part of discussion: 1) case assessment results; 2)assessment methods; 3) assessment tools.
Thank you for your suggestion, which we have carefully considered. We believe the current format assists the reader to see how the barriers, enablers and ERIC strategies are reflected in the broader body of evidence.

Round 2
Reviewer 3 Report
The manuscript has been improved well.